# Stage 5 Chronic Kidney Disease: Epidemiological Analysis in a NorthEastern District of Italy Focusing on Access to Nephrological Care

**DOI:** 10.3390/jcm13041144

**Published:** 2024-02-18

**Authors:** Francesca K. Martino, Giulia Fanton, Fiammetta Zanetti, Mariarosa Carta, Federico Nalesso, Giacomo Novara

**Affiliations:** 1Nephrology, Dialysis, Transplantation Unit, Department of Medicine (DIMED), University of Padova, 35124 Padua, Italy; federico.nalesso@unipd.it; 2International Renal Research Institute Vicenza, 36100 Vicenza, Italy; fntngiulia@gmail.com (G.F.); zanetti.fiammetta@gmail.com (F.Z.); 3Department of Laboratory Medicine, San Bortolo Hospital, 36100 Vicenza, Italy; mariarosa.carta@aulss8.veneto.it; 4Department of Surgery, Oncology and Gastroenterology, Urology Clinic University of Padua, 35124 Padua, Italy

**Keywords:** end-stage kidney disease, prevalence, mortality, elderly

## Abstract

Background: We conducted a retrospective epidemiological study about the prevalence of stage 5 chronic kidney disease (CKD) in a high-income district, comparing some demographic characteristics and outcomes of those patients who had nephrological consultations and those who had not. Results: In a district of 400,000 adult subjects in 2020, 925 patients had an estimated glomerular filtration rate (eGFR) under 15 mL/min and CKD. In the same period, 747 (80.4%) patients were assessed by nephrologists, while 178 (19.6%) were not. Age (88 vs. 75, *p* < 0.0001), female gender (66.3% vs. 47%, *p* < 0.001), and eGFR (12 vs. 9 mL/min, *p* < 0.001) were significantly different in the patients assessed by a nephrologist as compared those who did not have nephrological care. Furthermore, unfollowed CKD patients had a significantly higher death rate, 83.1% versus 14.3% (*p* < 0.0001). Conclusions: About 20% of ESKD patients did not receive a nephrologist consultation. Older people and women were more likely not to be referred to nephrology clinics. Unfollowed patients with stage 5 CKD had a significantly higher death rate.

## 1. Introduction

In the last few years, high-income countries reported end-stage kidney disease (ESKD) prevalence around 800–2400 per million population [1,2,3,4]. A 2016 meta-analysis of the prevalence of chronic kidney disease (CKD) in the world suggested a higher prevalence of CKD in higher-income countries such as Europe, the USA, and Canada. In this context, ageing is strongly associated with CKD, with an estimated prevalence of CKD of about 28% in patients over 70 years [1]. Furthermore, other factors such as higher prevalence of hypertension, obesity, diabetes, smoking habits, and female gender strongly impact CKD and ESKD occurrence [1]. The U.S. Renal Data System 2020 Annual Data Report corroborated this trend, documenting a rise in the incidence of CKD over the last 20 years. Specifically, ESKD patients increased from 97,856 to 134,837, and the prevalence of diabetes and heart failure in the older class of patients also rose [2,3]. Similarly, other countries documented a higher diffusion of ESRD, such as the United Kingdom, Germany, France, Portugal, and South Korea, with a prevalence between 1.9 and 3.5% in CKD adult patients [4]. It is interesting to notice the significant discrepancy in the prevalence of ESKD between high-income and low-income countries. The disparity in detecting CKD could partially explain this gap, considering the difference in resources associated with per capita healthcare expenditures [4]. A Korean report emblematically showed that the number of dialysis patients increased sixfold between 2002 and 2017, from 11,215 to 67,486, with a contextual growth of age from 55.57 (±13.31) to 62.13 (±13.23) years [5]. Worldwide, 2.5 million people have ESKD, with significant disparities in treatment. In Western countries, most of them receive hemodialysis, peritoneal dialysis, and kidney transplants, while a significant number of them do not receive adequate treatment in African, Eastern European, and Latin American countries. As reported by Freedberg et al., the availability of treatment impacts patient survival, increasing the mortality rate, which seems to reach 30% in ESKD patients in lower-income countries [4].

The growth of the elderly population in higher-income countries could be bound to several factors, including quality of life improvement at the home and workplace and easier access to medical care related to per capita healthcare spending [4]. The nephrologists’ perception moves in this direction, considering an increasing number of elderly patients in dialysis units and the type of outpatients. In the last decades, there has been an increase in CKD secondary only to diabetes, hypertension, and vascular disease [6]. Over time, the etiopathogenesis of CKD has changed. Specifically, more and more commonly, ESKD patients are elderly [1,3], with a history of previous acute cerebrovascular accident, ischemic heart disease, or neoplastic disease, and survived thanks to healthcare improvements. Over time, we observed an eight- to ten-fold increase in patients treated by renal replacement therapy, considering the prevalence of dialysis at around 100–300 per million in the eighties [7]. If the longer life expectancy of dialysis patients partially justifies this increase, the higher incidence of ESKD is the main factor impacting the current prevalence [1,3,8]. The increasing number of dialysis patients could also be related to the limited expansion of conservative management in older people [9]. Theoretically, ESKD patients can more easily receive proper treatment in high-income countries, especially where the health system is public. Specifically, the public health system should provide adequate care for all patients without discrimination related to personal wealth. As reported by Ghazi et al., a patient’s socioeconomic status and type of healthcare are independently associated with CKD prevalence regardless of the patient’s age [10]. However, socioeconomic status can strongly impact lifestyle [11,12] and disease awareness, especially in older people [13]. Theoretically, more accessible access to nephrological care without cost should reduce care disparity, at least in the more symptomatic phase of CKD. In stage 5 CKD, specialistic management is mandatory to correct sodium and potassium imbalance, mineral bone disease, metabolic acidosis, and anaemia and initiate replacement therapy. The lack of nephrologist assistance and the absence of correction of traditional and non-traditional cardiovascular risk factors harm patient survival, as reported by Milkowski et al. [14].

The detection of CKD seems related to health system organization and could differ throughout geographic areas. Specifically, there is an annual estimation of dialysis prevalence in Italy, but it still lacks an estimation of stage 5 CKD. Previous studies, such as INCIPE [15], MATISS [16], the VIP study [17], and the Sardinia Cohort study [18], reported a general prevalence of CKD. However, they referred to the 1990–2010 period without stage 5 CKD assessment. Knowledge about the gap between effective prevalence and actual access to nephrologist care in stage 5 CKD patients is essential for managing the expansion of ESKD in terms of facilities and personnel resources. Furthermore, it also seems fundamental to catching and correcting possible healthcare inequity.

The present study aims to estimate the actual prevalence of ESKD in one high-income area with a public health system, focusing on the prevalence of patients who received treatment from nephrologists and those without nephrological care.

## 2. Materials and Methods

We performed a retrospective epidemiological study to assess the prevalence of stage 5 CKD in a northeastern Italian district. The study was conducted at the Nephrology and Laboratory Medicine departments of San Bortolo Hospital, Vicenza, Italy, from 1 January 2020 to 31 December 2020.

We conducted the study following the Declaration of Helsinki. Ethical review and approval were waived for this study as required for the retrospective clinical investigation; we informed the local ethics committee protocol ESRD.PREVALENCE n.14/22. Every patient was associated with a unique code to permit the identification of each medical record and save patient privacy.

The patients who performed serum creatinine level tests in the hospital laboratory were eligible for the study. Inclusion criteria were patients ≥18 years old, with at least one estimated glomerular filtration rate (eGFR) <15 mL/min and at least another eGFR under 20 mL/min during the period between 3 and 12 months before the fall of eGFR at stage 5 CKD [19]. The exclusion criteria were:-Belonging to other healthcare system districts;-The presence of acute kidney injuries with the subsequent improvement of eGFR over 30 mL/min.

In the evaluated district, the national healthcare system employed all nephrologists, who usually perform their activity in public hospitals, making the number of private outpatient clinics negligible. Nephrologists registered patients’ reports in a computerised medical system. Furthermore, in our district, blood examinations are mainly performed by the hospital laboratory. In this context, we can estimate eGFR in all people with a creatinine test and assess who had a nephrologist’s care.

Specifically, we created a patient code comprising the first three letters of the surname, the first three letters of the first name, followed by 1 for woman and 2 for man, six numbers for the birthdate, and the last three characters of the national healthcare code. First, we detected all patients with eGFR under 15 mL/min and matched each patient with their code. Contextually, we caught all patients with at least one outpatient clinic visit in the examined period and linked each patient with their code. Therefore, we crossed the nephrological computerised medical records with the records of the hospital’s laboratory to discover the prevalence of followed and unfollowed stage 5 CKD patients.

We considered only age and gender for each subject and, eventually, the type of nephrology outpatient clinic associated with the patient.

We estimated eGFR by CKD epi formula [8]: eGFR = 142 * min(standardised Scr/K, 1)α * max(standardised Scr/K, 1) − 1.200 * 0.9938Age * 1.012 [if female], where Scr (serum creatinine) = mg/dL; K = 0.7 (females) or 0.9 (males); α = −0.241 (females) or −0.302 (males); min = indicates the minimum of Scr/K or 1; max = indicates the maximum of Scr/K or 1.

Creatinine was measured using the enzymatic method with an automatic analyzer (Dimension Vista, Siemens Healthcare, Tarrytown, NY, USA).

Finally, we recorded the patient’s death in the first year after the blood examination.

### Statistical Analysis

We represented continuous and categorical variables as median values with interquartile range (IQR) and proportion (%), respectively. We compared followed and unfollowed patients by Mann–Whitney-U for continuous variables such as age and eGFR, while using Chi-square tests for categorical variables such as gender and death. A two-tailed *p* < 0.05 was considered statistically significant.

The statistical analyses were performed using IBM SPSS Statistics, version 27 (IBM Corp., Armonk, NY, USA).

## 3. Results

In our district, during 2020, our healthcare system followed about 400,000 adult subjects, including 1044 patients with at least one eGFR under 15 mL/min. Among those 1044 patients, 119 subjects were excluded. Specifically, 102 patients had acute kidney injury (AKI), and 17 came from another district. Finally, 925 patients had ESKD, according to the laboratory results (around 2310 per million population), including 747 patients (80.4%) who were followed by a nephrology department (1867 per million population), including 349 who had renal replacement therapy (872.5 per million population). The number of nephrologists between 1 January and 31 December 2020 was fourteen.

Figure 1 reports the distribution of ESKD subjects considering the nephrology follow-up and type of nephrology clinic. Table 1 reports age and gender in the entire population and all subgroups (followed, unfollowed, haemodialysis patients, peritoneal dialysis patients, kidney transplant patients, late referral, and outpatients).

Interestingly, the proportion of patients who were not followed in the nephrology department was around 19.6%. Those patients were significantly older (*p* < 0.0001), had significantly higher eGFR (*p* < 0.001), and were more frequently women (*p* < 0.001), as compared to those followed by nephrologists. Figure 2 reports histograms for age and eGFR considering patients who had and who did not have nephrologist care.

On the whole, 252 CKD patients died, of whom 107 were followed by a nephrologist (14.3%), and 148 did not receive any nephrologist care (83.1%) (*p* < 0.001). Specifically, regarding unfollowed patients, the number of women alive in one year after blood examination was 27 (22.8%), while the number of men was 8 (13.3%) (*p* = 0.1).

## 4. Discussion

Our analysis showed a prevalence of 2300 ESKD per million people in a northeastern area of Italy. About 80% of the patients received at least one nephrologist consultation, while about 20% did not receive any specialistic evaluation and treatment. We observed how patients without nephrologist care were more commonly elderly and female, with higher eGFR and an much higher incidence of death. Overall, we showed a significant gap between the nephrologist assessment and treatment of stage 5 CKD patients and its prevalence in the general population in Italy. Our results corroborate the previous results regarding the prevalence of ESKD, which was estimated to be between 800 and 2400 per million people in different high-income countries [4]. They are consistent with the 2018 USRDS registry, which reported 2242 ESKD cases per million people adjusted for age, race, and sex [3]. Moreover, our estimation of renal replacement therapy is congruent with the figures of several other studies [1,4,5,8]. Currently, the prevalence of ESKD patients who do not receive haemodialysis remains poorly evaluated in Europe [20]. It is almost obscured in Italy, where the primary studies about CKD prevalence did not report ESKD prevalence [15,16,17,18]. The lack of previous reports in Europe, especially Italy, does not allow for any comparison with the present analysis data.

We recorded a high prevalence of stage 5 CKD in elderly patients. Those results could be related to the higher prevalence of classic risk factors of CKD, such as diabetes, hypertension, and cardiovascular disease [21,22]. Furthermore, the ageing of the general population [23] and the physiological changes in renal function by ageing [24], such as nephron reduction, glomerular sclerosis phenomenon, reduction of kidney reserve function, and tubular impairment, could exacerbate chronic kidney failure in the oldest people. Finally, the improvement in survival for dialysis patients can explain the remarkable prevalence of older patients. Specifically, we found untreated patients were significantly older than those with nephrologist consultation. Kidney failure in older people is usually associated with other comorbidities such as dementia, cognitive impairment [25,26], chronic heart failure, cardiovascular disease [27,28], diabetes, and hypertension [1,2,29], which dominate the clinical picture in ageing people. CKD is an indolent disease, which can limit patients’ awareness [30,31,32] and, likely, general practitioners’ diagnoses. The rate of mortality in the whole population (27.6% of ESKD) is consistent with the data from the USRDS registry from 2018 [33], which reported 310 deaths for 1000 ESKD white patients over 75 years.

The difference in the mortality rate between followed and unfollowed patients (14.3% vs. 83.1%, *p* < 0.001) was impressive and merits a careful interpretation considering the lack of information about comorbidities. Such a remarkable gap could be partially related to the basal conditions of patients and partially to the lack of adequate treatment in old and frail patients. Currently, it is difficult to hypothesise which condition is prevalent. Future studies are necessary to corroborate our results and detect the impact of comorbidities and nephrological care on patient survival. A capillary campaign of sensitization of geriatricians, palliative care physicians, and general practitioners should be addressed to increase ESKD awareness among such physicians. Further strategies, such as the implementation of telemedicine, a more intensive collaboration with nephrologists, the opportunities for nephrologists to visit patients at home, and the set-up of a multidisciplinary team could improve awareness of nephrological care and, consequently, patient outcomes.

We recorded a higher prevalence of women with ESRD who did not have any nephrologist care. Our finding confirmed the results of Stolpe et al., who described lower awareness in female patients in stage 4 CKD in Germany [31]. Ours and the German results move in the same direction as the National Health and Nutrition Examination Survey (NHANES), which reported a higher awareness about CKD in men between 1999 and 2018. The gap of awareness between genders has been reducing in the last four years of observation and seems more prevalent in the Caucasian race [33]. It is challenging to understand which conditions can lead to gender disparity in CKD awareness. However, considering the interquartile age range in unfollowed patients of 82–91 years, this difference could have roots in the disparity in school education and social roles between men and women born between 1930 and 1940. We did not find a significant difference related to gender disparity in the patient’s survival, although the *p* value of our analyses (equal to 0.1) seems to reflect a different trend in the mortality rates between men and females. Furthermore, considering female distribution in outpatient clinics, we corroborate the results of the Australian report that showed a higher inclination to choose conservative management in the female gender, confirming a gender difference in treatment selection [34].

Our results show an impressive number of ESKD elders in unfollowed patients, who could receive, for example, conservative management. Unfortunately, to date, this therapeutic option seems underused despite its possible positive influence on elderly and comorbid patients, allowing effective control of uremia, delaying CKD progression and the need for dialysis, maintaining their habits, and limiting hospitalization without heavy effects on the quality of life [9]. Likely, a campaign of information on conservative management among general practitioners and geriatrics could positively impact the care of older and frail patients. Finally, the number of nephrologists in the examined district is slightly higher than in other Western countries, which varies between 20 and 30 pmp [4]. The number of nephrologists per million people can be considered as a proxy for the attitude of each area to supply adequate nephrological care such as dialysis, kidney transplant, and the use of erythropoiesis-stimulating agents, calcium mimetics, potassium, and phosphate binders. Our results suggest a grey area in the healthcare system in higher-income areas, with about 20% of patients who did not have any consultation with nephrologists despite the high number of nephrologists in the area.

The present analysis has some limitations. Firstly, we did not evaluate comorbidities such as diabetes, chronic heart failure, cardiovascular disease, cognitive impairment, chronic pulmonary disease, and tumours. Consequentially, it is hard to take univocal advice from our findings, and we discourage interpreting our results to mean that the lack of assessment for ESKD was the primary cause of death. As mentioned above about the death rate, it is reasonable to think that the choice not to refer older patients to nephrologists could be the consequence of their poor health conditions. However, on the other hand, specialist consultation and optimised nephrology treatment could ameliorate the patients’ conditions and improve their prognoses. Secondly, we did not have eGFR from the private laboratories in our district. This issue could eventually underestimate the number of ESKD patients in our analysis. Thirdly, some patients in our district could have performed blood examinations in other districts and been followed by local nephrologists. This event can theoretically underestimate the number of followed CKD patients. Those phenomena are likely marginal in our district and generally in Italian regions because ESKD patients are usually managed in the nearest center to limit the time necessary to reach the hospital and because supportive pharmacological therapy such as erythropoietin, phosphorus binding, and potassium binding can be prescribed only by public health system nephrologists. Furthermore, in frail and elderly people, general practitioners require blood samples at home to limit the inconvenience related to the transport to the hospital. Finally, we cannot exclude a reduction in nephrologist care related to the COVID-19 pandemic, especially in the first four months of 2020, when there was a lockdown in Italy. Considering the need for mandatory treatment in stage 5 CKD patients regarding erythropoietin, phosphorus and potassium binders, and renal replacement therapy, it is unlikely that there will be a prolonged delay of over one year. Future research about stage 5 CKD is needed to confirm those findings in other contexts, hopefully including race, comorbidities, socioeconomic issues, and treatment in the research features. Our study had limited access to the national system database to evaluate comorbidity, school education, and economic conditions. Despite the limitations of information about comorbidities and socioeconomic features, we believe our report remains relevant because it uncovers a failure in nephrological care, which is the first step to improving assistance for stage 5 CKD patients.

## 5. Conclusions

Our report observed that nephrologists never saw a significant proportion of ESKD patients and that those patients were more commonly octogenarian and female and experienced a poor prognosis.

## Figures and Tables

**Figure 1 jcm-13-01144-f001:**
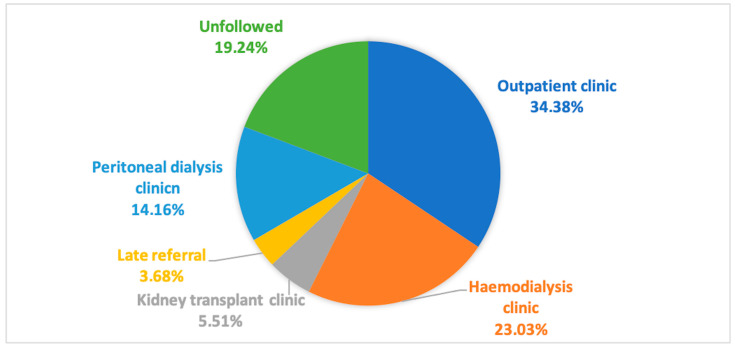
ESKD patients according to nephrology care.

**Figure 2 jcm-13-01144-f002:**
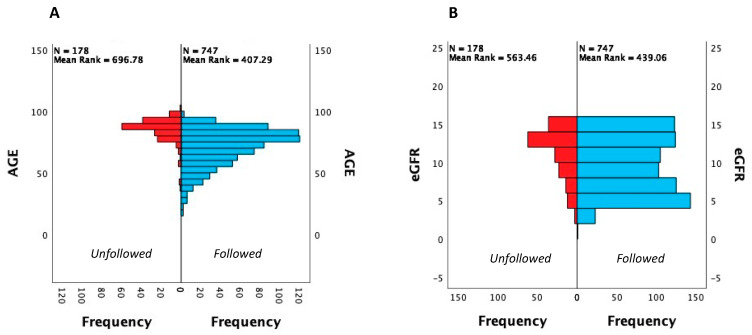
Age (**A**) and eGFR (**B**) distribution in followed and unfollowed ESKD patients.

**Table 1 jcm-13-01144-t001:** Age and gender in the entire population and all subgroups.

	All Population (925)	Followed (747)	Unfollowed (178)	Haemodialysis Clinic (213)	Peritoneal Dialysis Clinic (131)	Kidney Transplant Clinic (51)	Outpatient Clinic (318)	Late Referral (34)	*p* Values
Age, yrs (median, IQR)	78 (65.5–86)	75 (62–83) *	88 (82–91) *	70 (61–78)	64 (53–76)	51 (41–62)	82 (75–87)	75.5 (68–83)	* <0.0001
Female gender (%)	50.7%	47% *	66.3% *	41.8%	35.1%	44.7%	56.6%	47.1%	* <0.001
eGFR, ml/min (median, IQR)	10 (6–13)	9 (6–12)	12 (9–13) *	6 (5–8)	6 (5–9)	8 (5–10.5)	12 (10–14)	8.5 (5–10.2)	* <0.001
Death (%)	27.6%	14.3% *	83.1% *	16.9%	9.9%	10.5%	14.5%	20.6%	* <0.0001

Note: * refers to the comparison groups (followed and unfollowed patients) and corresponding *p*-value.

## Data Availability

Data available on request due to restrictions, e.g., privacy or ethical. The data presented in this study are available on request from the corresponding author.

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
