# Peer review of "Stage 5 Chronic Kidney Disease: Epidemiological Analysis in a NorthEastern District of Italy Focusing on Access to Nephrological Care"

_jcm, 2024, doi:10.3390/jcm13041144_

Round 1

Reviewer 1 Report

Comments and Suggestions for Authors

Martino et al. present a retrospective cohort study focussing on the characteristics of individuals with end-stage renal disease within a district in Italy.

I have the following comments to make:

Title. Needs to be improved. Doesn't inform the reader about the type of study being conducted and the subject area isn't clear too.

Line 30. Isn't a combination of factors responsible for the perceived increased prevalence of CKD, including better detection and awareness. I get the impression this sentence attributes CKD just to aging and this is erroneous and simplistic. The sentences in lines 31-35, seem to contradict the first statement. 

Line 55, Dyslipidaemia is not a cause of CKD but is associated with it.

Lines 57-58. Poorly worded sentences. What is "survived ictus"?

Lines 67-68. Why are your aims important? What makes this study important or potentially impactful?

Line 100-101. This sentence doesn't make much sense i.e. "type of nephrology ambulatory associated..."

 Line 161. Prevalence of CKD where specifically? In Italy?

Line 171. I cannot understand how you justify aging to CKD stage V. Alone it clearly doesn't result in CKD stage V. Indeed, it is still debatable whether can  the aging kidney represents CKD. It clear that other more influential factors are contributory such as co-morbidities including hypertension and diabetes, polypharmacy, and dehydration. Although you acknowledge this to some extent it should be mentioned first. Also these factors are not supported by your dataset  - a major limitation

Line 183. The finding of elderly females with CKD5 who had high rates of mortality is interesting, but little has been discussed on why this was the case in your population. Further data should have been examined in this sub-cohort. Could have the COVID19 pandemic played any role here? It might be helpful to examine whether such trend was noticeable in previous years - this would be a piece of further research work.

Line 193-194. The number of nephrologists in the geographical area (14) examined in this study is not comparable in my view to other Western populations (20-30).

Lines 201-218. Although the limitations of this study are pointed out, little is suggested on how they could be mitigated in future study. There isn't much discussion either on future directions of study,

Comments on the Quality of English Language

Needs improvement overall. 

Author Response

Dear Reviewer,

Thanks for your punctual observations of the manuscript. We find your suggestions smart and really helpful. After your revision we appreciated an improvement of our manuscript, so we hopefully attend your feedback.

Specifically:

  • Needs to be improved. Doesn't inform the reader about the type of study being conducted and the subject area isn't clear too.

We changed the title as required in " Stage 5 chronic kidney disease: epidemiological analysis in a north-eastern district of Italy focusing on the access to nephrological care."

  • Line 30. Isn't a combination of factors responsible for the perceived increased prevalence of CKD, including better detection and awareness. I get the impression this sentence attributes CKD just to aging and this is erroneous and simplistic. The sentences in lines 31-35, seem to contradict the first statement.

We revised all paragraph.

  • Line 55, Dyslipidaemia is not a cause of CKD but is associated with it.

We revised the phrase

  • Lines 57-58. Poorly worded sentences. What is "survived ictus"

We revised all paragraph.

  • Lines 67-68. Why are your aims important? What makes this study important or potentially impactful?

We add a new paragraph to better explain the aim of the study and its impact.

  • Line 100-101. This sentence doesn't make much sense i.e. "type of nephrology ambulatory associated..."

We changed as required.

  • Line 161. Prevalence of CKD where specifically? In Italy?

We added the request information and appropriate reference

  • Line 171. I cannot understand how you justify aging to CKD stage V. Alone it clearly doesn't result in CKD stage V. Indeed, it is still debatable whether can the aging kidney represents CKD. It clear that other more influential factors are contributory such as co-morbidities including hypertension and diabetes, polypharmacy, and dehydration. Although you acknowledge this to some extent it should be mentioned first. Also these factors are not supported by your dataset - a major limitation

We revised the paragraph to better explain the possible factors related to CKD stage 5. Thanks for your advice.

  • Line 183. The finding of elderly females with CKD5 who had high rates of mortality is interesting, but little has been discussed on why this was the case in your population. Further data should have been examined in this sub-cohort. Could have the COVID19 pandemic played any role here? It might be helpful to examine whether such trend was noticeable in previous years - this would be a piece of further research work.

We revised the paragraph and we compare our results with other report. Specifically, about Covid 19 pandemic we are agree about the possible interference of pandemic in our report, so we add a new paragraph in the discussion, unfortunately we can able right now to compare 2020 with previous or followed year. Thanks for your suggestion.

  • Line 193-194. The number of nephrologists in the geographical area (14) examined in this study is not comparable in my view to other Western populations (20-30).

We chanced the phrase; over all you are wright 34 is slightly higher than 20-30.

  • Lines 201-218. Although the limitations of this study are pointed out, little is suggested on how they could be mitigated in future study. There isn't much discussion either on future directions of study,

We add a new paragraph about that. Thanks again for your suggestion.

Reviewer 2 Report

Comments and Suggestions for Authors

The manuscript titled "End-stage kidney disease: concern about the Prevalence and Outcome of patients not seeking nephrological care in a high-income area of Italy" presents a retrospective analysis of the demographics and outcomes of end-stage kidney disease patients in a northeastern Italian district, noting marked disparities in age, gender, and kidney function between those who received nephrological care and those who did not. Despite these findings, the study falls short in its academic contribution as it merely presents data without delving into the causative factors or offering practical recommendations for improving patient outcomes. In its present form, the study does not provide the robust analysis or innovative insights.

Author Response

Dear Reviewer,

Thanks for your observations of the manuscript.

Our job is interesting for nephrologists, geriatricians, and general practitioners as the first step to discovering the possible gap between nephrologist perception and the actual prevalence of ESKD and its risk factors of iniquity. Furthermore, to the best of our knowledge, the job's novelty is analysing nephrological care access in an entire area in a class of patients who surely need nephrological care. However, we are also aware of the limitations of our report, such as the lack of comorbidities, race, and socioeconomic issue detection, which make it difficult to provide general practical recommendations to improve care. Our first aim was to detect the presence of an issue with nephrological care access; after assessing the disparity, we can explore the risk factor and the possible solution with a project. Based on your comment, we analysed the associated factors and got some recommendations in the discussion.

We hope you can appreciate an improvement.

Reviewer 3 Report

Comments and Suggestions for Authors

The authors examined the prevalence and outcomes of end-stage kidney disease (ESKD) patients in a region of Italy. Specifically, it compared ESKD patients who received nephrology care to those who did not. In 2020, there were 925 ESKD patients (eGFR <15 ml/min) in a region with 400,000 adults. Of those, 747 patients (80.4%) were seen by nephrologists while 178 patients (19.6%) did not receive nephrology care. The patients without nephrology care were significantly older (median age 88 vs 75 years), more likely to be female (66.3% vs 47.0%), and had higher eGFRs (12 vs 9 ml/min) compared to nephrology patients. Strikingly, ESKD patients without nephrology care had an 83.1% mortality rate over the study period, compared to only 14.3% in those receiving care. About one-fifth of ESKD patients in this region did not see a nephrologist. These patients tended to be older, female, and have higher eGFRs. However, they had dramatically worse outcomes, with over 80% dying within a year. This highlights a concerning gap in care for some ESKD patients. Increased awareness and nephrology referrals may improve outcomes in this population.

  1. The authors did not evaluate key comorbid conditions like diabetes, heart failure, cardiovascular disease, cancer etc. Adding data on comorbidities would better characterize the ESKD populations, risk factor profiles, and allow adjustment for confounding factors. Specifically, they should report prevalence of various comorbidities in the nephrology care and non-nephrology groups and include them as covariates in multivariate regression models exploring predictors of mortality.
  2. The study relied only on eGFR data from the hospital lab to determine ESKD prevalence. Expanding data collection to include private/outpatient labs could provide a more accurate estimate of true community ESKD rates and reduce potential underestimation from sole reliance on hospital lab GFR testing.
  3. Some ESKD patients may have received nephrology care outside the examined health district, which could make prevalence of "unfollowed" patients appear higher falsely. The authors should make efforts to access medical records across adjoining districts through regional health information exchanges or other means to clarify if patients received outside nephrology care.
  4. The single health district examined may limit generalizability of findings. Expanding this to a multi-center study across diverse geographic regions and care settings would improve external validity and better inform nationwide policies.
  5. They did not report details on renal replacement modality (in-center hemodialysis, home dialysis, transplant etc). Incorporating treatment specifics could reveal clinical insights into how mortality differs across modalities.
  6. Only simple univariate statistical tests were conducted without adjusting for potential confounders in the comparison groups. Applying more advanced multivariate regression techniques would allow adjustment for comorbidities, demographics, renal function etc. to better isolate the impact of nephrology care on mortality.
  7. No data presented on why patients failed to receive nephrology referrals or the barriers involved. Surveying primary care providers and reviewing charts on documented reasoning behind no specialist referral could provide vital information to improve processes.
  8. Conservative kidney management is mentioned but frequency and associated outcomes not examined. Reporting detailed rates of conservative management vs. renal replacement therapy could better inform care decisions, especially for elderly patients.

Author Response

Dear reviewer,

Thank you for your suggestions and advice about our report. We know its limits, as reported in a detailed paragraph in the discussion. However, our report remains relevant because it showed a gap in nephrological care, which is the first step to improving assistance in stage 5 CKD patients.

Specifically,

  1. We reported in the limits of the study the lack of comorbidities detection. Detecting comorbidities would make our job more informative, but we need more resources and access to the national system database. Our manuscript reported significant findings, i.e., the gap between the nephrologist's perception of stage 5 CKD and the reality.
  2. The expansion of data collection, including private labs, would be desirable, but obtaining it requires other personal resources. We underlined this issue as a limitation of our study. As reported in the discussion, the missing data from private labs could only underestimate the rate of unfollowed patients with stage 5 CKD.
  3. In our area, stage 5 CKD patients are followed mainly by local nephrologists for different reasons:
  • Priority to have the outpatient clinic, blood examination, and radiological scan is valid only in the district.
  • The frequency of visits is often high, and the patients generally prefer local places for care.
  • Places for dialysis are always limited, so patients are referred to local services. In any case, this issue was mentioned in our study's limitations.

4. As reported in the discussion, other studies should corroborate our findings.

5. The study aimed to detect the prevalence of stage 5 CKD. Other studies thoroughly evaluated the mortality rate in followed patients, considering the type of renal replacement therapy.

6. Considering the context of our report (age, gender, nephrological care, death), we believe the univariate analysis is enough. Our report aims to estimate the actual prevalence of ESKD, focusing on nephrological care. We appreciate suggestions, and we will consider them in future research

7. More information about patients and barriers would make our job more informative. The idea is attractive but challenging to achieve, considering the study's retrospective nature, the patients' ages, and the mortality rate.

8. In 2020, conservative management experienced limited numbers, less than ten patients in the examined district. A robust conservative management program was born in March 2021, so we cannot draw any conclusions about that.

Reviewer 4 Report

Comments and Suggestions for Authors

Authors show in this manuscript that about 20 % of patients with advanced CKD was not evaluated by a nephrologists. Moreover, patients with uncontrolled ESKD had higher risk of death compared to patients receiving specialist care.  

However:

1) please use arabic numbers when mentioning CKD stages, according to KDIGO guidelines;

2) please do not repeat words, when already abbreviation was introduced, i.ex. CKD/chronic kidney disease (L55);

3) please use more specific/medical words, not 'ictus', 'heart attack', 'tumour' (L57-58);

4) please be more specific about 'creatinine blood examinations', rather 'serum creatinine level test/analysis';

5) please use appropriately abbreviation of CKD-EPI equation (L102);

6) please check the manuscript, some data seem to be misplaced (i.ex. '80.4' in line 120);

7) Figure 1: please correct caption = 'ESKD patients (type/visits) according to/IN the nephrology care';

8) especially please check the discussion section, 'dementia' is not a 'cognitive disorder'? (L177), 'heart failure' and 'hypertension' a 'cardiovascular disease'? (L177-178);

9) if higher female prevalence in your study may be related with lower CKD awareness, did it have an impact on patients survival? how many males and females died in your groups?

10) please omit words like 'nephrologist never saw... a patients' (L220-221).

Comments on the Quality of English Language

Some minor changes are needed (i.ex. L20) and more medical definitions/style of writing will improve the manuscript.

Author Response

Dear Reviewer,

Thanks for stimulating comments of the manuscript, and especially for the interest in our job. We find suggestions well placed and really helpful. The revision improves the level of our manuscript, so we hopefully wait for your feedback.

Specifically:

  • please use arabic numbers when mentioning CKD stages, according to KDIGO guidelines;

We revised all manuscript about this point.

  • please do not repeat words, when already abbreviation was introduced, i.ex. CKD/chronic kidney disease (L55);

We revised all manuscript about this point.

  • please use more specific/medical words, not 'ictus', 'heart attack', 'tumour' (L57-58);

We changed as appropriate.

  • please be more specific about 'creatinine blood examinations', rather 'serum creatinine level test/analysis';

We changed as appropriate.

  • please use appropriately abbreviation of CKD-EPI equation (L102);

We changed as appropriate.

  • please check the manuscript, some data seem to be misplaced (i.ex. '80.4' in line 120);

We rephrase the sentence as required.

  • Figure 1: please correct caption = 'ESKD patients (type/visits) according to/IN the nephrology care';

Thanks, we did.

  • especially please check the discussion section, 'dementia' is not a 'cognitive disorder'? (L177), 'heart failure' and 'hypertension' a 'cardiovascular disease'? (L177-178);

We rephrase the sentence as required.

  • if higher female prevalence in your study may be related with lower CKD awareness, did it have an impact on patients survival? how many males and females died in your groups?

We did the analyse of survival according to gender and we add a paragraph. Thanks for your suggestion.

  • please omit words like 'nephrologist never saw... a patients' (L220-221).

We changed the phrase.

Reviewer 5 Report

Comments and Suggestions for Authors

The report delves into the prevalence and outcomes of end-stage kidney disease (ESKD) patients who forgo nephrological care, emphasizing the significance of nephrological intervention even in affluent regions. The findings hold implications for healthcare strategies, emphasizing the imperative for heightened awareness among healthcare providers. While the study lacks an evaluation of comorbidities that could significantly influence mortality rates and omits data from private laboratories, its publication is warranted to enhance awareness within the healthcare community.

I have a few comments for the authors to address:

The study does not specify the duration of follow-up for the included patients.

In the Statistical Analysis subsection, consider swapping the first two sentences to enhance clarity. It is currently unclear what Mann-Whitney-U and Chi-square tests are being applied to, respectively.

The paper could provide further elaboration on the implications of the findings for healthcare policies and potential interventions to address identified gaps in nephrology care seeking. Discussing how these results could inform future healthcare strategies would enhance the practical significance of the study.

Thorough proofreading is recommended to address potential grammatical errors, typographical mistakes, or issues related to punctuation and syntax. Additionally, review the text for any unnecessary repetition of information or ideas.

Comments on the Quality of English Language

The recommendations encompass addressing grammatical errors, enhancing sentence structure, introducing more transition phrases, and ensuring consistent verb tenses to improve overall clarity and coherence.

Author Response

Dear Reviewer,

Thanks for your punctual observations of the manuscript, and thanks especially for understand the spirit and importance of our study, considering the limitations of our study . We find your suggestions well placed and really helpful. After your revision we appreciated an improvement of our manuscript, so we hopefully attend your feedback.

Specifically:

The study does not specify the duration of follow-up for the included patients.

We reported this important feature in the methods. Thanks to notice this information gap.

In the Statistical Analysis subsection, consider swapping the first two sentences to enhance clarity. It is currently unclear what Mann-Whitney-U and Chi-square tests are being applied to, respectively.

We better describe the type of test to compare each variable.

The paper could provide further elaboration on the implications of the findings for healthcare policies and potential interventions to address identified gaps in nephrology care seeking. Discussing how these results could inform future healthcare strategies would enhance the practical significance of the study.

We added more one paragraph about the possible strategies. Thanks for your suggestion.

Thorough proofreading is recommended to address potential grammatical errors, typographical mistakes, or issues related to punctuation and syntax. Additionally, review the text for any unnecessary repetition of information or ideas.

We revised potential grammar and graphical mistakes. Thanks for your advice.

Round 2

Reviewer 1 Report

Comments and Suggestions for Authors

I agree with the changes the authors have made.

Comments on the Quality of English Language

Now reads better.

Reviewer 2 Report

Comments and Suggestions for Authors

No further comment.

Reviewer 4 Report

Comments and Suggestions for Authors

Authors corrected the manuscript as requested.